# Blood Plasma Proteome: A Meta-Analysis of the Results of Protein Quantification in Human Blood by Targeted Mass Spectrometry

**DOI:** 10.3390/ijms24010769

**Published:** 2023-01-01

**Authors:** Anna A. Kliuchnikova, Svetlana E. Novikova, Ekaterina V. Ilgisonis, Olga I. Kiseleva, Ekaterina V. Poverennaya, Victor G. Zgoda, Sergei A. Moshkovskii, Vladimir V. Poroikov, Andrey V. Lisitsa, Alexander I. Archakov, Elena A. Ponomarenko

**Affiliations:** 1Institute of Biomedical Chemistry, 119121 Moscow, Russia; 2Federal Research and Clinical Center of Physical-Chemical Medicine, 119435 Moscow, Russia; 3Department of Biochemistry, Medico-Biological Faculty, Pirogov Russian National Research Medical University, 117997 Moscow, Russia

**Keywords:** proteome, targeted mass spectrometry analysis, human proteome project, knowledge databases

## Abstract

A meta-analysis of the results of targeted quantitative screening of human blood plasma was performed to generate a reference standard kit that can be used for health analytics. The panel included 53 of the 296 proteins that form a “stable” part of the proteome of a healthy individual; these proteins were found in at least 70% of samples and were characterized by an interindividual coefficient of variation <40%. The concentration range of the selected proteins was 10^−10^–10^−3^ M and enrichment analysis revealed their association with rare familial diseases. The concentration of ceruloplasmin was reduced by approximately three orders of magnitude in patients with neurological disorders compared to healthy volunteers, and those of gelsolin isoform 1 and complement factor H were abruptly reduced in patients with lung adenocarcinoma. Absolute quantitative data of the individual proteome of a healthy and diseased individual can be used as the basis for personalized medicine and health monitoring. Storage over time allows us to identify individual biomarkers in the molecular landscape and prevent pathological conditions.

## 1. Introduction

Blood biochemistry tests have long been used to assess the human health status. The complete set of plasma proteins (proteome) plays a special role in diagnosing socially significant diseases, including cardiovascular diseases, diabetes mellitus, and different types of cancer. Researchers have long been interested in answering questions of what proteins are present in blood plasma, what their concentrations are, and how these plasma proteomes differ between a healthy and sick individual [1]. The use of monoclonal antibodies is currently the gold standard for detecting and quantifying proteins in blood plasma. This approach has several drawbacks, including its limited specificity owing to the cross-reactivity of antibodies and low potential for multiplex assays. Qualitative and quantitative assessments of protein markers in blood plasma will significantly improve diagnostic efficiency. Currently, SOMAscan and Olink are prevalent affinity-based techniques that have excellent potential for multiplexity and are often preferred over mass spectrometry [2,3].

Shotgun and targeted mass spectrometry approaches, whose development has been rapidly advancing, are excellent alternatives to antibody-based techniques for studying the plasma proteome. High-resolution shotgun mass spectrometry allows the simultaneous study of several hundreds of proteins with high specificity. Approximately 3000 proteins have been detected using this technology under the HUPO Human Plasma Proteome Project (HPPP) owing to the joint efforts of 35 laboratories; 900 of these proteins have been identified with high confidence. More than 1500 proteins have been detected in blood plasma at concentrations ranging from 10^−12^ to 10^−3^ M [4]. This observation provides grounds for considering blood plasma as a complex biological matrix. In 2005, the research team led by Aebersold demonstrated the important role of meta-analysis, that is, summarizing the results of several experiments to obtain a complete proteomic map. Different proteomic techniques, sample preparation options, and algorithms for analyzing the resulting data lead to the identification of different sets of proteins in the same sample [5]. In 2011, as part of creating the Peptide Atlas resource, this research team compiled a list of 1929 master proteins identified in blood plasma with high confidence using LC-MS/MS [6,7]. According to the Human Protein Atlas, 4072 proteins have been identified over a broad concentration range in the plasma of healthy people using mass spectrometry without stable isotope labeling [8].

Targeted methods, including selected/multiple reaction monitoring (SRM/MRM), enable the detection of proteins in the biomaterial and quantification of their absolute concentrations, which is critical for medical applications. Using synthetic isotope-labeled peptide standards as reference samples improves the identification accuracy and measured protein concentration [9]. Furthermore, SRM analysis can be used to detect proteins present at low (10^−14^–10^−12^ M) and ultralow (10^−18^–10^−15^M) concentrations via the irreversible binding of minor plasma proteins to biogranules [10] or fractionation [11,12,13].

Fewer studies on blood plasma proteins have utilized the SRM method compared to shotgun analysis. Hüttenhain et al. [14] used the SRM method to investigate candidate biomarkers for malignant neoplasms. They identified 182 plasma proteins, presumably associated with tumor diseases, with spectral counts as their primary characteristic. Domanski et al. [15] employed SRM to design a panel consisting of 67 potential marker proteins for diagnosing cardiovascular diseases. The normal concentrations of the proteins estimated using various methods, including mass spectrometry, were obtained from the literature.

As the SRM method enables structural analysis of the amino acid sequence of peptides, it is increasingly used to detect point amino acid substitutions in proteins. Thus, this method can be used to quantify proteoforms [16] or validate mRNA editing events by ADAR enzymes at the protein level [17].

SRM technology has entered the global market as multiplex test kits for research. Such kits are manufactured by Biognosys AG (Schlieren, Switzerland) and Cambridge Isotope Laboratories (Tewksbury, MA, USA) (PeptiQuant kit) [18]. The evolution of these test kits involves increasing the number of detectable analytes and improving analysis sensitivity, thus making them an efficient tool for protein marker detection. This method can quantify up to 500 proteins per sample; however, the reference concentrations in these kits are not provided to users.

The next logical step is to adapt targeted mass spectrometry for clinical applications. Designing and improving the available SRM test kits can enable the personalized prediction of disease risks, diagnosis, and assessment of treatment efficacy based on rational drug prescription and individual treatment selection to generate human digital molecular images [19]. A combination of labeling and targeted MS in the SRM mode is currently widely used as an alternative to immunoassays for accurate protein quantification and confirmation of the findings of earlier studies where biomarkers had a controversial clinical significance [16].

Identifying the reference ranges for analyte proteins is an important challenge for introducing SRM test kits in clinical practice. For example, such information is not available for the Biognosys AG kits. In the SRM studies performed by Hüttenhain et al. and Domanski et al., the normal concentrations of the proteins under study were obtained from the literature; the reference values were obtained using different methods, including those unrelated to mass spectrometry [14,15]. Notably, the differences between the methods employed for collecting serum or plasma from patients (e.g., using test tubes coated with different anticoagulants) may lead to statistically significant variations in protein levels, thereby impeding biomarker research [20].

In the present study, a set of reference concentrations of proteins that could be used for SRM analysis was generated according to a meta-analysis of the SRM data for plasma collected from healthy volunteers. Furthermore, the fundamental principles for selecting potential biomarker candidates and designing panels for human health monitoring were formulated.

## 2. Results and Discussion

### 2.1. Meta-Analysis of the SRM Data for Plasma from Healthy Volunteers

Publications for the meta-analysis of the quantitative proteomics data were selected using two strategies. The ScanBious platform [21], which allows classification of research publications from the PubMed database according to keywords and the selection of semantically similar publications based on the integrated semantic similarity analysis system, was employed for the first strategy. Figure 1 shows the search results obtained with the query *“Selected (or Multiple) Reaction Monitoring Proteomics Human”*.

As the internal search algorithm data across the PubMed database included enrichment with synonyms, the retrieved information included data from both SRM and MRM analyses. The list included 197 entries on a specified topic published over the past ten years. The literature primarily comprised studies using the SRM method to measure the concentration of single proteins. Kumar et al. [22] showed that the plasma concentration of apolipoprotein F in healthy individuals was 445.1 ng/mL, with a coefficient of variation (CV) <12%. In many studies, this method was used as a supplementary technique for validating active pharmaceutical substances in human plasma [23,24,25] and assessing pharmacokinetics [26]. After 2012, the MRM analysis began to be used as a method that allows targeted quantification of plasma proteins, including proteoforms encoded by a single gene (e.g., quantification of the Apoe3 and Apoe4 isoforms differing by one or two amino acid residues [16]). Therefore, these methods are currently used more frequently for detailed investigations of individual proteins and their proteoforms than for quantitative proteome screening, which is promising for medical applications. Nonetheless, this screening in the gene-centric format was proposed as the experimental method of choice for implementing the International Human Proteome Project, which is of fundamental significance for the evolution of medicine [27].

The second strategy for selecting research publications involved the biocuration of PubMed literature sources. Based on these findings, researchers have been selecting the SRM/MRM method for proteomic analysis less frequently (Figure 2a). One plausible reason is that other methods, such as tandem mass tag (TMT) labeling, which enables multiplex investigation of the samples and determination of the relative and absolute peptide concentrations, are now commonly used [28,29]. Moreover, affine protein enrichment with aptamers on the SOMAscan platform has been implemented, which enables the simultaneous detection of several hundred to thousand proteins [30]. The quality of shotgun proteomics data quantification can also be improved using the data-independent acquisition (DIA) method instead of data-dependent acquisition (DDA) [31].

The comparison of SRM and aptamer-based methods is of particular interest, as aptamer-based methods have been promoted in recent years as a powerful alternative to the “gold standard” antibody-based approach. Table 1 presents the important features of the SOMAscan platform and SRM assays.

A comprehensive comparison revealed that the SRM method has advantages, such as high specificity, sensitivity, low reagent development cost, absolute quantitation capability, and amino acid sequence structural analysis capability. However, rapid analysis of as many individual proteomes as possible is crucial for clinical applications. In this regard, SOMAscan platforms that employ high-throughput microarrays are still more desirable than the relatively slow MRM method. However, problems have recently been observed with aptamer-based cross-platform correlation [36]. Further developments in mass spectrometry, such as ultrafast analysis [37], hold promise for the retention of mass spectrometry on the “waiting list” as an alternative to antibody-based methods.

Finally, despite its high selectivity and sensitivity, the evaluation of SRM data is only particularly automated compared to shotgun proteomics methods. Careful visual assessment of the spectra by qualified personnel is required [38]. 

As a descending trend (see Figure 2a) exists for the application of the SRM method, most datasets obtained using this method with the current level of analytical sensitivity have already been published and are available in databases [13].

**Figure 2 ijms-24-00769-f002:**
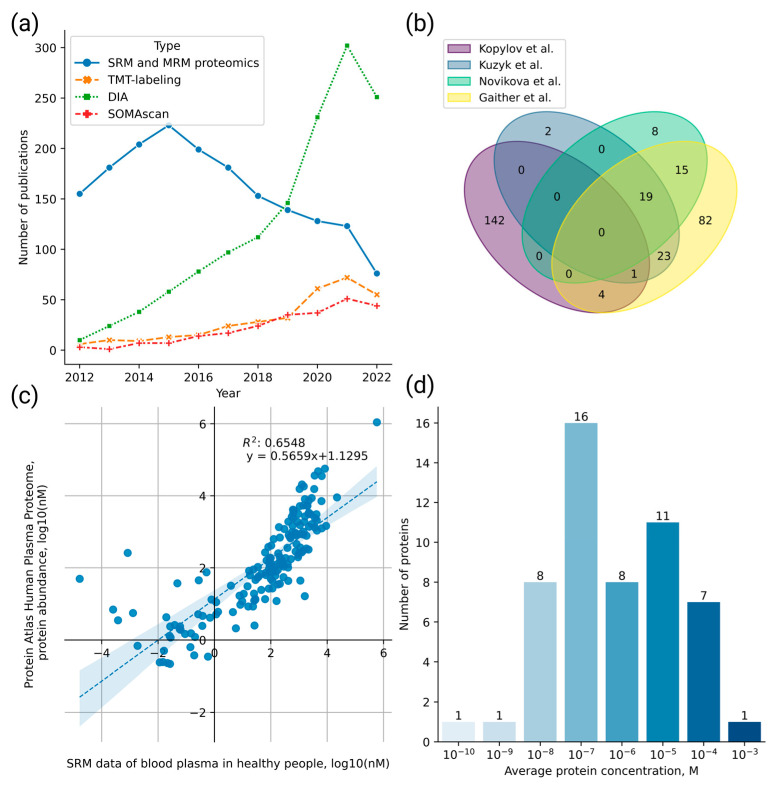
Meta-analysis of the SRM data for plasma from healthy volunteers. (**a**) Comparison of the number of published studies employing different proteomic methods in the PubMed database over the past ten years. A descending trend for selected and multiple reaction monitoring over the past years is highlighted in blue. (**b**) Venn diagram based on a comparison of the datasets obtained using the SRM method by Kopylov et al., Novikova et al., Kuzyk et al., and Gaither et al. [18,33,35,39]. (**c**) Correlation between protein concentrations (Log 10 (nM)) measured in the plasma of healthy volunteers using the SRM method in four studies [18,33,35,39] and concentrations of the same proteins reported in the Human Protein Atlas database. (**d**) Range of protein concentrations detected in more than 70% of samples and characterized by the interindividual coefficient of variation <40% obtained via a meta-analysis of studies [18,33,35,39].

Therefore, in the present study, a meta-analysis of publications was performed to generate a summary table of the plasma protein concentrations in healthy volunteers based on the SRM method. The already-known but insufficiently annotated peptide concentrations that vary in the plasma of healthy volunteers can be helpful in further research and searches for biomarkers of various diseases.

Following a review of the literature sources and criteria for dataset inclusion in the meta-analysis, we selected publications reporting the absolute concentrations of at least ten proteins measured in the plasma of healthy volunteers. The results of the meta-analysis of the data obtained by quantitative SRM screening of blood plasma from healthy volunteers are reported in a tabular format (Appendix A), specifying protein identifiers in the UniProt database, the names of the protein and its encoding gene, the average concentration, and the CV characterizing interindividual variability.

Among all the analyzed publications, only four datasets obtained using the SRM method with synthetic isotope-labeled peptide standards met the specified criteria. Protein concentrations in these studies were obtained by measuring a single synthetic peptide. The first dataset reported by Kopylov et al. [35] contained data on the concentrations of 147 peptides (measured in 54 samples), whose genes belong to chromosomes 18 and 13, and the Y chromosome and mitochondrial DNA. A second publication by Novikova et al. [33] presented data on the concentrations of 42 peptides (measured in 31 samples) belonging to FDA-approved proteins. The third dataset was obtained from Kuzyk et al. [39]. The measured concentrations of 45 proteins in 20 plasma samples were previously reported. The fourth publication by Gaither et al. [18] reported quantitative data on the maximum and minimum concentrations of 144 proteins measured in at least five of the 20 analyzed commercial human plasma samples and one pool. EDTA was used as an anticoagulant to obtain plasma in all studies. Therefore, the comprehensive list contained concentrations of 296 proteins, 20 of which were shared by three datasets, and 42 by two datasets (Figure 2b). By comparing the absolute concentrations of the shared proteins, we identified the maximum and minimum values and calculated their fold changes. For 55 proteins, the range of concentration variations was within one order of magnitude. However, the concentrations of proteins, such as antithrombin III, alpha-2-macroglobulin, complement component C3, transthyretin, transferrin, and apolipoprotein B100, differed by a factor of 10 or more. The comparison results are presented in Appendix A.

The final list containing 296 proteins was compared with the data for 4072 proteins from the Human Protein Atlas project obtained via mass spectrometry based on open-source Peptide Atlas project data. Two datasets shared the 172 values shown in Figure 2c, and the values shown in Appendix A were shared by the two datasets. As shown in Figure 2c, the correlation between the values decreased significantly only at low concentrations (≤10^−12^ M for the Human Protein Atlas project data). The medium and high concentrations (from 10^−11^ to 10^−5^ M)) measured according to the number of spectral identifications were comparable to the results obtained using the SRM method. The Human Protein Atlas project data can be used to design the diagnostic panels.

To create the overall list reported herein (Appendix A), we selected the values with the highest CV showing interindividual variability when concentrations were available from at least two sources. Data with high CV values were obtained for numerous samples and may provide a more realistic assessment of the value when selecting potential biomarkers. To identify the proteins with the most stable concentrations, CVs were used to show interindividual differences. We measured the percentage of samples containing a particular protein whenever possible. Proteins found in at least 70% of the analyzed samples with an interindividual CV <40% were considered as a stable characteristic of the proteome of a healthy human.

In the resulting list, only 53 proteins met the formulated criteria and could be considered a stable characteristic of the proteome of a healthy human (Appendix A). Figure 2d illustrates the range of the protein concentrations. The highest concentration was in the range of 10^−8^ to 10^−4^ M. The remaining 243 proteins measured using the SRM method in the four analyzed datasets were examined as a variable group.

Biological data annotation was used to analyze the enrichment of the most stable proteins (core proteins) and variable proteins according to the categories in the Gene Ontology (GO) database (Figure 3).

The analysis demonstrated that in terms of their cell localization, the most stable proteins (core proteins) are components of chylomicron with high confidence (FDR q-value = 3.1 × 10^−3^) (Figure 3A). According to the GO category “Biological Processes,” stable proteins were involved in lipid metabolism regulation (Figure 3C), and their primary function was binding receptors of lipoprotein particles (FDR q-value = 2.57 × 10^−2^).

For the group of variable proteins consisting of 243 proteins, along with cellular localization, proteins were identified to be primarily located in chylomicrons (Figure 3B), and to participate in organophosphate ester transport, lipoprotein particle remodeling (Figure 3D), and in the activity of phosphatidylcholine-sterol O-acyltransferase activator and anion binding (FDR q-value = 1.18 × 10^−2^ and 2.07 × 10^−2^, respectively).

Data on the enrichment of a stable set of proteins with the lipoprotein fraction can be used to design panels for monitoring lipid metabolism in healthy humans and individuals with a high risk of atherosclerosis (e.g., genetic predisposition, obesity, hypertension, and smoking history).

### 2.2. Panel for Monitoring Human Health Status and Potential Biomarker Selection

Here, we sought to demonstrate how data on the healthy human plasma proteome can be evaluated in the context of diseases. The list of the most stable proteins was annotated using the DisGeNET platform (Figure 4), the largest database of gene–disease associations [40]. The 30 proteins found to be associated with different diseases are listed in Appendix A.

Figure 4 indicates that enrichment was more significant for the set of stable proteins than for variable proteins. Thus, an association was revealed between some stable proteins, hereditary systemic amyloidosis, and type IV hyperlipoproteinemia, which are rare familial disorders. The concentration variation for the set of stable proteins revealed in this study may be rare and may require further study. Proteins with variable concentrations were primarily associated with cardiovascular and inflammatory diseases (Appendix A).

The set of proteins with the most stable concentrations can be used to identify associations with various familial disorders. The composition of potential protein panels designed for MRM studies can be easily modified (e.g., by adding or replacing the protein analytes). Owing to such plasticity, researchers can employ the MRM technology to perform test repositioning (similar to drug repositioning) using well-studied and diagnostically relevant protein analytes in new combinations, which enables the assessment of the biological status for which they have never been used before.

The results of the meta-analysis of protein concentrations in the blood of healthy individuals were compared to the measured concentrations of the same proteins in human plasma for two diagnosed diseases. We selected data obtained using the SRM method for plasma samples collected from patients with cancer and neurological diseases. The first dataset comprised the protein concentrations in plasma samples collected from patients with neurological diseases [42]. The second dataset for comparative analysis was obtained from a study that focused on biomarkers of lung adenocarcinoma [43]. The results are presented in Table 2.

As shown in Table 2, proteins with the most stable concentrations in healthy volunteers differed from those in patients with a pathology by no more than two orders of magnitude. The maximum difference (1.7-fold) was demonstrated for alpha-2-macroglobulin, a component of the innate immune system; variation in its concentration was detected earlier in individuals with neuronal damage [44,45]. A comparison of the remaining values in Table 2 revealed that the concentration of ceruloplasmin (*CERU*) was lower by almost three orders of magnitude in patients with neurological diseases than in healthy volunteers. This protein is known to regulate iron efflux, whereas ceruloplasmin deficiency may cause iron overload and can be associated with neurodegenerative disorders [46,47]. Notably, the concentrations of gelsolin isoform 1 (*GSN*) and complement factor H (CFH gene) were significantly reduced in patients with lung adenocarcinoma. Gelsolin is an actin-binding protein that affects cell mobility and maintains cytoskeletal integrity. The expression of gelsolin is variable in some cancer cell lines and malignancies, such as non-small-cell lung and breast cancer [48]. Complement factor H, a component of the complement system, plays a crucial role in tumor immune surveillance [49]. Recent studies have demonstrated that CFH can play a pivotal role in the resistance to complement-mediated lysis in different cancer cells, including lung cancer cells [50]. Detection of such drastic variations in concentrations can be an incentive for further research.

For proteins not shown with color in Table 2, concentrations differed by 1–1.5 orders of magnitude. For most of them (displayed in bold), concentration values were also most stable in the plasma of healthy volunteers. Therefore, the data listed in Table 2 and Appendix A can be used to select potential single proteins or entire protein arrays for designing diagnostic panels. If the differences in concentrations between the samples of healthy individuals and patients with pathology are minimal, another protein should be selected for further research.

## 3. Materials and Methods

Publications for the analysis were selected using the ScanBious platform [21], which allows publications to be retrieved from the PubMed database upon inquiry and visualizes related keywords as a semantic network. Semantically similar studies can be classified based on an integrated semantic similarity analysis system.

The data from the Human Protein Atlas project [8] obtained using mass spectrometry, available as open-source data of the Peptide Atlas project, were used to compare the absolute protein concentrations. GO enrichment analysis was performed using the GOrilla tool [51].

An analysis to determine the enrichment of the most stable and variable proteins with molecular components associated with the diseases was performed using the Enrichr module of the Gseapy library (v. 0.13.0) across categories of the DisGeNET database (https://www.disgenet.org/, accessed on 10 November 2022); the *p*-value cutoff was <0.05 [51]. The ten most robust categories for each protein group (stable and variable proteins) were visualized.

## 4. Conclusions

Absolute quantitative proteomic data are essential for future biomarker discovery and the creation of personal digital molecular images for healthy and sick people. Information on the quantitative proteome of the blood plasma of healthy people will generate a standard for comparison with the proteome of a diseased person, significantly expanding the possibilities of diagnostics via biological markers.

Currently, proteomic methods have limitations in sensitivity, that is, the MS-detector is triggered in the presence of at least one million molecules in 1 µL of sample (10^–12^ M). Perhaps in the future, analytical proteomics may allow us to detect and measure one molecule in 1 μL (10^–18^ M), which will significantly increase the number of analyzed proteins in the framework of health monitoring.

Estimating the ranges of interindividual and individual variability will enable the transfer of medical approaches to personalized diagnostics, risk assessment of diseases, and therapy. In addition to the early diagnosis of diseases and evaluation of treatment efficacy, such data can serve as a basis for identifying new drug targets. Although the trend of targeted proteomics methods is declining, the use of such methods has resulted in the accumulation of an impressive quantitative data array for various human tissues and organs and an understanding of the difference between proteomic landscapes of healthy and diseased persons. These data can be used for cross-validation of the results obtained with new, more sensitive methods and can become the basis for the formation of hypotheses for subsequent studies. In the present study, we selected a dataset of human plasma quantitative measurements to provide researchers with reference values for protein concentrations. Until enough experiments are accumulated using new methods, SRM data will serve as the optimal basis for estimating variability and reliability. We believe that the efforts of the global community will be directed toward the development of cheaper methods with better analytical sensitivity and reproducibility.

## Figures and Tables

**Figure 1 ijms-24-00769-f001:**
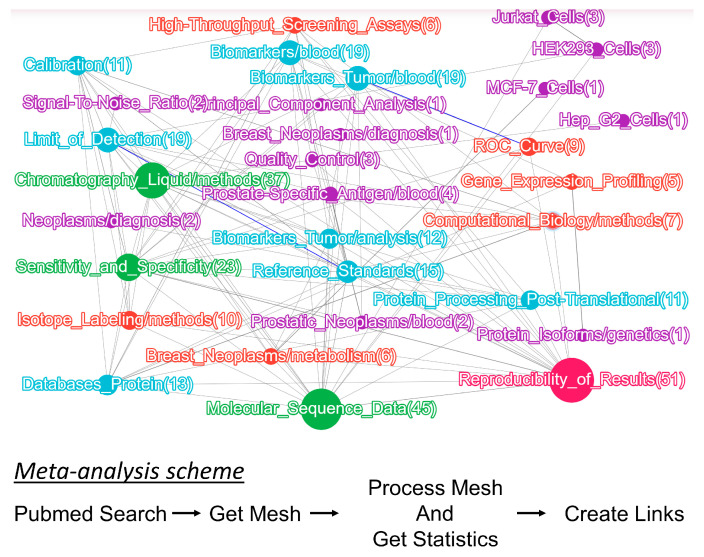
Results of the search for publications reporting the selected/multiple reaction monitoring (SRM/MRM) data for humans over the past ten years (2012–2022) using the ScanBious system. The size of the nodes reflects the occurrence of the keywords in the abstracts. The identified key clusters involve search terms related to tumor biomarkers, their sensitivity, specificity, and reproducibility. Of note, SRM analysis was frequently used to study cancer cell lines, such as Jurkat, MCF-7, and HepG2.

**Figure 3 ijms-24-00769-f003:**
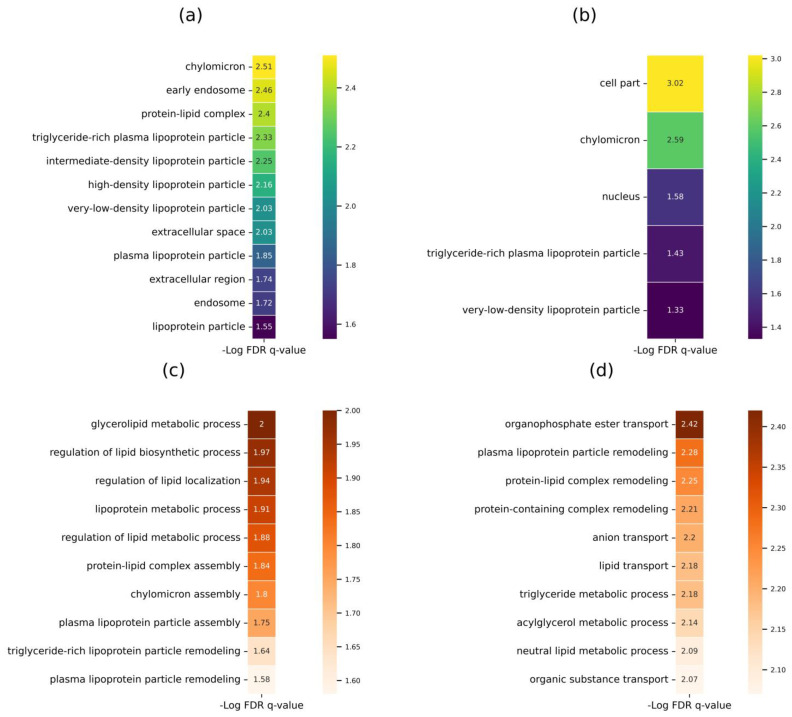
Gene Ontology enrichment analysis of the lists of respective proteins obtained using the SRM method. (**a**) Core proteins (*N* = 53) belonging to the GO category “Cell Component”, (**b**) Variable proteins (*N* = 243) belonging to the GO category “Cell Component”, (**c**) Core proteins (*N* = 53) belonging to the GO category “Biological Processes”, (**d**) Variable proteins (*N* = 243) belonging to the GO category “Biological Processes”. Core proteins (the most stable ones) are the proteins found in more than 70% of the samples with CV <40%. Variable proteins are those found in less than 70% of the samples or have a CV >40%.

**Figure 4 ijms-24-00769-f004:**
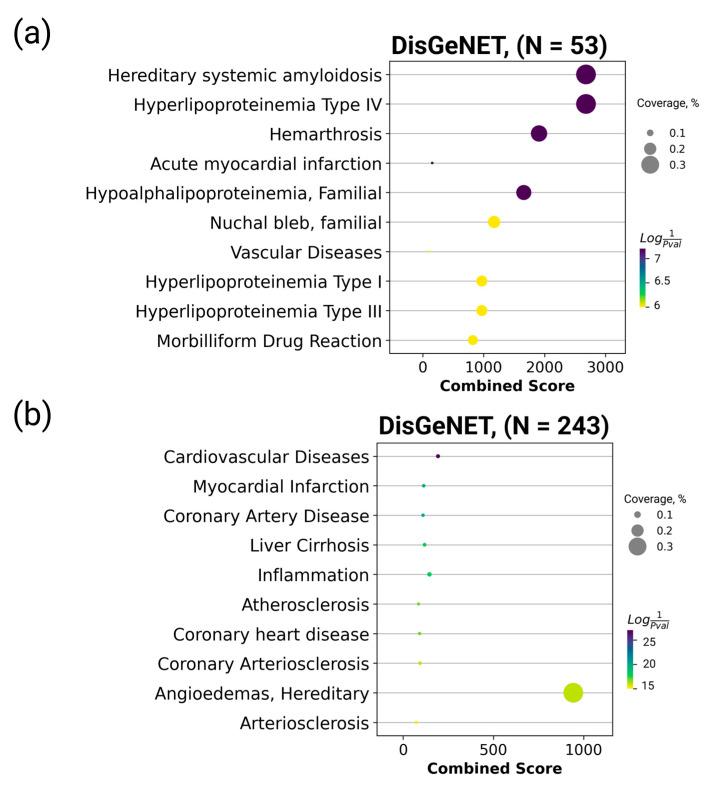
Enrichment analysis of stable proteins found to be associated with diseases according to the DisGeNET database obtained for (**a**) the list of the most stable proteins and (**b**) the variable dataset. The combined score is the decimal logarithm of the *p*-value determined using the Fisher’s test and multiplied by the z-score [41]. “Coverage, %” is the percentage of target protein associated with this disease.

**Table 1 ijms-24-00769-t001:** Comparison of the mass spectrometric SRM (selected reaction monitoring) approach and aptamer-based SOMAscan platform for protein measurement in human serum/plasma.

Method Feature	SOMAscan	SRM
Volume of crude serum/plasma, µL	15–50 [2,32]	2.5–5 [33]
Sample preparation	Not needed	Tryptic digestion
Special reagent	DNA-based aptamers	SIS peptides
Multiplexity of protein analytes	813–7288 [2,32,34]	111–329 [33,35]
Quantification type	Relative [36]	Absolute
Direct/indirect protein quantification	DNA serves as an intermediary	Peptide amino acid sequence serves as an intermediary
Sensitivity	10^−14^–10^−3^ [32]	10^−16^–10^−3^ [34,35]
Dynamic range in serum/plasma, orders of magnitude	8–10 [34]	4–8 [34,35]
High throughput array multiplexing	Yes	No
Structure analysis, analysis of SAP, isoforms, etc.	No	Yes [17]

**Table 2 ijms-24-00769-t002:** Comparison of protein concentrations in healthy individuals and patients with pathologies obtained using the multiple reaction monitoring method. The cells where protein concentrations differed by more than two orders of magnitude are shown in blue. The most stable proteins in the plasma of healthy volunteers retrieved via meta-analysis are displayed in bold [42,43].

Protein Name	Gene Name	Log 10 (Average Concentration, fM, M × 10^−15^)		
		Healthy Human Plasma	Neurological Diseases (*n* = 19, Kiseleva et al., Clin Trans Med, 2015) *	Lung Adenocarcinoma (*n* = 102, Wu et al., Proteomics Clin Appl, 2020)
**Alpha-1-antitrypsin**	** *A1AT* **	**10.0**	**9.5**	
**Alpha-2-macroglobulin**	** *A2MG* **	**10.2**	**8.5**	
**Apolipoprotein A-I**	** *APOA1* **	**10.6**	**9.1**	
*Ceruloplasmin*	*CERU*	*9.1*	*6.5*	
**Complement C3**	** *CO3* **	**9.2**	**8.4**	
Cystatin-C	*CST3*	7.7	7	
**Fibrinogen alpha chain**	** *FIBA* **	**10.3**	**8.9**	
Haptoglobin	*HPT*	9.9	9	
**Hemopexin**	** *HEMO* **	**9.4**	**8.3**	
Insulin-like growth factor-binding protein 3	*IGFBP3*	7.4	7	
**Plasma protease C1 inhibitor**	** *IC1* **	**8.6**	**7.5**	
Platelet basic protein	*CXCL7*	8.0	6.8	
**Serotransferrin**	** *TRFE* **	**10.3**	**9.3**	
Serum albumin *	*ALB*	11.8	10	
**Transthyretin**	** *TTR* **	**9.9**	**8.5**	
**von Willebrand factor**	** *VWF* **	**7.8**	**7**	
*Complement factor H*	*CFH*	*9.0*		*5.8*
**Desmoglein-2**	** *DSG2* **	**7.0**		**6.4**
Gelsolin, isoform 1	GSN	9.2		6.8
Lambda-crystallin homolog	*CRYL1*	6.6		6.2
Lumican	*LUM*	8.5		6.7
Mucin-16	*MUC16*	7.7		5.8

* The average concentrations calculated using the data of three protocols were used for the dataset in the publication by Kiseleva et al. Albumin concentration was determined using the values obtained for the protocol without depletion.

## Data Availability

Data sharing not applicable. No new data were created or analyzed in this study.

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
