# Peer review of "Blood Plasma Proteome: A Meta-Analysis of the Results of Protein Quantification in Human Blood by Targeted Mass Spectrometry"

_ijms, 2023, doi:10.3390/ijms24010769_

Round 1

Reviewer 1 Report

In their manuscript entitled "Blood plasma proteome: a meta-analysis of the results of protein quantification in human blood by targeted mass spectrometry" Kliuchnikova et al., summarize proteomic data, providing absolute protein concentrations to lead the scientific community towards a reference value-based plasma protein biomarker research. The manuscript is well written and in this Reviewer's opinion it is of great value for the scientific community.

The only (minor) comment I have is that the conclusion section reminds me of an abstract and vice versa. So I would suggest reconstructing the abstract and the conclusion section. The abstract in my opinion needs more specific data and the conclusion section needs to mainly highlight the importance of this study and its future impact/potential in biomarker discovery.

Author Response

Reviewer:
In their manuscript entitled "Blood plasma proteome: a meta-analysis of the results of protein quantification in human blood by targeted mass spectrometry" Kliuchnikova et al., summarize proteomic data, providing absolute protein concentrations to lead the scientific community towards a reference value-based plasma protein biomarker research. The manuscript is well written and in this Reviewer's opinion it is of great value for the scientific community.
The only (minor) comment I have is that the conclusion section reminds me of an abstract and vice versa. So I would suggest reconstructing the abstract and the conclusion section. The abstract in my opinion needs more specific data and the conclusion section needs to mainly highlight the importance of this study and its future impact/potential in biomarker discovery.

Authors:
Dear Reviewer, we highly appreciate your in-depth analysis and substantial time looking over the manuscript. We wish to thank you for the constructive comment, which provided valuable insights to refine the contents and the style of our manuscript. We have significantly changed the abstract and the conclusion of the manuscript, taking into account your recommendations. We hope we made more useful for the readers.

Reviewer 2 Report

In this manuscript, titled "Blood plasma proteome: a meta-analysis of the results of protein quantification in human blood by targeted mass spectrometry" the authors attempt to provide the reference standards for a set of plasma proteins that can be used for the clinical evaluation of patients.

The manuscript is well-written and well presented. However, it would benefit from another round of editing to clarify the flow of reading at places (for example, line 125) and another reading to use the appropriate words (eg: line 151).

I congratulate the authors for identifying a crucial area which could be significantly improved by further research. However, I am concerned about the novelty of the study and it's use therefore. Though the authors set out to provide the perfect pitch for the study, they limit themselves to a method which is declining in it's applicability. The manuscript ends up being a review of just 4 other studies. The additional value this manuscript would provide as against the original manuscripts is not clear. This is a necessary point that this reviewer feels the need to be discussed.

The manuscript mentions the recent methods of analysis such as SOMAscan and Olink that have become much prevalent in their uses. However, there is no discussion of the advantages/disadvantages of the author-chosen method against these to provide the perspective of why the techniques of SRM and MRM are on the decline. This discussion, I believe would be of interest to the readers coming from the practitioners of the method.

Author Response

Reviewer:

In this manuscript, titled "Blood plasma proteome: a meta-analysis of the results of protein quantification in human blood by targeted mass spectrometry" the authors attempt to provide the reference standards for a set of plasma proteins that can be used for the clinical evaluation of patients. The manuscript is well-written and well presented. However, it would benefit from another round of editing to clarify the flow of reading at places (for example, line 125) and another reading to use the appropriate words (eg: line 151).

Authors:

To improve clarity of our writing we engaged a professional language editor. We have edited the text according to the Reviewers’ comments and recommendations provided by a language editor.

Reviewer:

I congratulate the authors for identifying a crucial area which could be significantly improved by further research. However, I am concerned about the novelty of the study and it's use therefore. Though the authors set out to provide the perfect pitch for the study, they limit themselves to a method which is declining in it's applicability. The manuscript ends up being a review of just 4 other studies. The additional value this manuscript would provide as against the original manuscripts is not clear. This is a necessary point that this reviewer feels the need to be discussed.

Authors:

Dear Reviewer, thank you for the opportunity to clarify the importance of our manuscript. Despite the fact that our results indicated the trend of targeted proteomics methods to decline, the SRM approach gave us an opportunity to accumulate an impressive quantitative proteomics data array for various human tissues and organs. These data can be used for cross-validation of the results obtained with new, more sensitive methods and become the basis for the formation of hypotheses for subsequent studies. In our manuscript, we have selected a data set of the human plasma quantitative measurements in order to provide interested researchers with reference values for protein concentrations. The final manuscript merges results from four articles, but in order to identify these four manuscripts, we had to process 197 studies. Similar datasets can be created for other popular types of biological material with potential clinical significance. Obviously, the efforts of the world community will be directed towards the development of cheaper methods with better analytical sensitivity and reproducibility. However, until enough experiments are accumulated using new methods, SRM data will be the optimal basis to estimate their variability and reliability.

Reviewer:

The manuscript mentions the recent methods of analysis such as SOMAscan and Olink that have become much prevalent in their uses. However, there is no discussion of the advantages/disadvantages of the author-chosen method against these to provide the perspective of why the techniques of SRM and MRM are on the decline. This discussion, I believe would be of interest to the readers coming from the practitioners of the method.

Authors:

The comparison of SRM and aptamer-based methods is of special interest, as the latter have been promoted in recent years as a powerful alternative to the “gold standard” antibody-based approach. To demonstrate the main features of the SOMAscan platform and SRM assays, we added table 1 to our manuscript. A comprehensive comparison shows that the SRM method has advantages such as high specificity, sensitivity, low reagent development cost, absolute quantitation capability, and amino acid sequence structural analysis capability. On the other hand, fast analysis of as many individual proteomes as possible is crucial for clinical application. In this regard, SOMAscan platforms that take advantage of microarray high-throughput are still more in demand than the relatively slow MRM-method. However, recently the problems with aptamer-based cross platform correlation have been observed.